# Eastern Arc of Glacial Relict Species—Population Genetics of Violet Copper *Lycaena helle* Butterfly in East-Central Europe

**DOI:** 10.3390/insects16121202

**Published:** 2025-11-26

**Authors:** Cristian Sitar, Marcin Sielezniew, Adam Malkiewicz, Zdenek Faltynek Fric, Martin Konvička, Hana Konvickova

**Affiliations:** 1Zoological Museum, Babes-Bolyai University, 5–7 Clinicilor, 400006 Cluj-Napoca, Romania; cristian.sitar@ubbcluj.ro; 2Emil Racovita Institute of Speleology, Clinicilor 5, 400006 Cluj-Napoca, Romania; 3Division of Biodiversity and Behavioural Ecology, Faculty of Biology, University of Bialystok, Ciołkowskiego 1J, 15-245 Białystok, Poland; marcins@uwb.edu.pl; 4Faculty of Biological Science, University of Wrocław, Przybyszewskiego 65, 51-148 Wrocław, Poland; amalki68@gmail.com; 5Institute of Entomology, Biology Centre CAS, Branišovská 31, 370 05 České Budějovice, Czech Republic; zdfric@gmail.com (Z.F.F.); hpatzenhauerova@gmail.com (H.K.); 6Faculty of Agrobiology, Food and Natural Resources, Czech University of Life Sciences, Kamýcká 129, 165 00 Praha, Czech Republic; 7Faculty of Sciences, University South Bohemia, Branišovská 31, 370 05 České Budějovice, Czech Republic

**Keywords:** butterfly conservation, glacial relic, interglacial habitats, introduced population, Carpathians, North European Plain

## Abstract

The Violet Copper (*Lycaena helle*) is distributed at high latitudes and among the mountains of Eurasia. It is protected in the EU, considered a glacial relic, and threatened by habitat loss and climate change. It has been studied mainly in Western Europe, where genetic studies revealed differentiation of population systems due to mutual isolation. We expand the knowledge of its population genetics into East-Central Europe, namely Poland (PL) and Romania (RO), where it inhabits both lowland fens and mountain grasslands, and we analyse the genetic makeup of a population established in the Czech Republic (CZ) by transfer of individuals from the Austrian Alps. We document processes of population differentiation, and also track the colonisation of the Northern European lowlands and high mountains, after recession of glaciers. Two populations inhabiting sparse broadleaved forests in Carpathian valleys may represent survival of the glacial cycles in situ and deserve the highest conservation priority.

## 1. Introduction

The current ranges of cold-adapted species of the Northern Hemisphere were established in response to rapid climate changes during the Pleistocene and the Holocene and the accompanying changes in vegetation and ecosystems [1,2,3]. During the last glacial period, southern regions of the glaciated North European Plain were covered by arid treeless steppe-tundra, also known as the mammoth steppe. The mammoth steppe biota combined cold-adapted elements currently associated with northern latitudes [4,5] with drought-adapted elements currently inhabiting the eastern Eurasiatic steppes [6,7]. This grasslands-dominated biome alternated with periglacial or glaciated mountains [8,9], and with woody plants refugia in sheltered south-exposed valleys [10] or other insular sites with warm mesoclimates [11]. Following the retreat of glaciers and the gradual colonisation of temperate and boreal regions by woody vegetation, species once associated with the steppe-tundra grasslands rescued themselves by migrating north or east, retreating to high altitudes, or survived in structurally suitable relic habitats [12], including azonal steppe grasslands in cases of drought-adapted species [13,14] or wetlands in cases of cold-adapted species [15,16,17]. Persistence of the azonal habitats into the period suitable for forest growth was facilitated by abiotic factors such as fires, activity of large herbivorous mammals and ultimately, the exploiting and management of the habitats by humans [18].

Species that were originally associated with the mammoth steppe and survived the Holocene while located in insular mountains and/or azonal refugia are also called “glacial relics” and may be particularly imperilled by ongoing climatic warming. A much-studied example of such species is the continentally endangered and EU-protected butterfly, the Violet Copper, *Lycanea helle* (Denis & Schiffermüller, 1775) (Lycaenidae). This humid grasslands-dweller is distributed broadly throughout the Palearctic, but is mostly found in isolated populations in the mountains of Western (from the Ardennes to Cantabrian Mts. and Pyrenees) and Central (Central Uplands and the Alps) Europe. Towards the northeast, scattered lowland populations exist from Mecklenburg–Western Pomerania to Poland, whereas in the southeast, there are few inhabited sites in the Carpathian Arc [19,20] and the Stara Planina Mts. [=Balkan Mts.] located at the Balkan peninsula [21,22]. Many lowland populations were lost during the 20th century due to habitat loss, and the butterfly became extinct, e.g., in the Czech Republic [23], and most of northeastern Germany [24]. The upland populations may suffer from over-intensive management, or management cessation followed by woody encroachment [25,26,27], while climate change represents the ultimate threat [27,28,29]. Notably, the mountain populations are univoltine [30], whereas the lowland populations form two to three generations per year [19,31,32]. The two situations are widely geographically separated in Western and Northern Europe [33] but may occur in proximity in the Carpathians (e.g., [19,20,33]).

During the last decades, *L. helle* was much studied and became one of the few European butterflies to which entire book was dedicated [34]. The studies revealed restricted dispersal [30,32,35], metapopulation dynamics [29,36], and dependency on advanced successional stages of humid grasslands or fens [19,37,38,39]. In terms of population genetics, the relic distribution in Western Europe manifests by strong differentiation among regional population clusters and high numbers of private alleles for individual regions, indicative of genetic drift [40,41]. Geographic differentiation appears weaker in the lowlands of Eastern Europe [33]. The conservative cytochrome oxidase I gene (the barcode), in contrast, does not exhibit any structure across Europe [42].

Listing of *L. helle* in the Habitats Directive raised concerns for its conservation [25,26,43], including population transfers. Descimon and Bachelard [44] describe two experimental introductions into unoccupied but ecologically suitable areas in France, the Morvan Mts. in 1975 and the Forez Mts. in 1982. The third transfer to an unoccupied area, carried out in 2001, directed from a population in the Türnitz Alps, a part of Northern Limestone Alps, Austria, to the Šumava Mts., the Czech Republic [39]. These three transfers allowed the rates of *L. helle* spatial expansion in unoccupied areas to be quantified.

This contribution expands upon the existing and impressive knowledge of *L. helle* population genetics by applying microsatellite markers, developed for use in earlier studies [40,45], to a selection of lowland populations in Poland, and both low- and high-altitude Carpathian populations in Romania. In addition, we analyse the situation of the newly established population in the Šumava Mts., the Czech Republic, which is linked with populations inhabiting the mountains of Western Europe. We target the following questions: (1) How do the genetic makeups of populations in the North European plains and the Carpathian Arc differ from each other, and how do they differ from the previously explored populations in the mountains of Western Europe? (2) What are the genetic links between the mountain and lowland populations in Romania, and how are their differences in voltinism reflected? (3) Did the Šumava transfer translate the entire genetic variability of its source Eastern Alps population?

## 2. Materials and Methods

### 2.1. Study Species

Within its large distribution area, *L. helle* inhabits a wide range of habitats, provided that its host plants, *Bistorta officinalis* Delarbre in boreal and temperate regions or *B. vivipara* (L.) in the Arctic [46], are present. In temperate Europe, most of the populations inhabit grasslands and fens with high cover of *B. officinalis*, a forb prospering in more advanced successional stages [26,47]. In addition, lowland populations inhabiting sparse deciduous forests exist in the wider Carpathian area [19] (Figure 1).

The univoltine mountain *L. helle* populations fly from late May to early July, whereas the lowland bivoltine populations form the first generation in April–early June, the second in July–early August, and occasionally, the third in late August–September [31,32,48,49]. Notably, lowland populations in northern Poland used to be univoltine and turned bivoltine during last half century [50]. Males employ perching to acquire females [19], both sexes use shrubs or small trees for overnight roosting [25,31]. Mobility is restricted, with maximum movements recorded < 1 km and males crossing shorted distances than females [30,35,38]. In newly established populations, the rate of expansion is 200–400 m per year [39]. Larval feeding lasts 4–6 weeks, pupation occurs in plant litter. The pupa is the overwintering stage.

### 2.2. Sampling and DNA Analysis

We sampled 161 *L. helle* individuals from ten populations (1 from the Czech Republic –CZ, 5 from Poland—PL, and 4 from Romania—RO) (Figure 1, Table 1). They represent all regions of PL inhabited by the butterfly, with a balanced representation of small, large, interconnected and isolated populations; all known populations in Romania; and the only population in Czechia. The sampled butterflies were stored in 96% ethanol until DNA extraction (Tissue Genomic DNA Mini Kit (Geneaid Biotech Ltd., Taiwan), in accordance with the manufacturer’s instructions, and the DNA was stored at −21 °C.

We characterised the populations by altitude, voltinism, area, and connectivity, with the latter two coded at simple ordinal scales (area: 1—units of ha, 2—tens to lower hundreds of ha, 3—higher hundreds of ha; connectivity: 1—isolated site, with no occupied or suitable sites up to 10 km, 2—occupied or suitable habitats within 10 km, 3—part of a metapopulation system with multiple occupied sites within a 5 km diameter).

We used the identical five microsatellite loci developed by [45], which facilitated direct comparison with the earlier results. The forward primers were fluorescently labelled (labels NED, VIC, FAM, PET). All loci were amplified in one PCR using the Multiplex PCR kit (Qiagen, Hilden, Germany). The concentration of primers varied from 0.2 to 0.5 μM. The conditions of touchdown PCR were as follows: initial denaturation at 95 °C (15 min), 10 cycles of 94 °C (30 s), 55 °C (90 s) and 72 °C (90 s), 26 cycles with T_a_ lowered to 52 °C with final elongation at 72 °C for 10 min. The capillary electrophoresis of obtained PCR products was conducted by Macrogen Europe BV (Amsterdam, The Netherlands). The genotypes from received electrophoretograms were scored using Geneious Prime 2025.1.2 (https://www.geneious.com).

The null allele frequency was checked using the programme FreeNA (https://www1.montpellier.inra.fr/CBGP/software/FreeNA/ accessed on 6 June 2025) [51], the linkage disequilibrium and Hardy–Weinberg equilibrium were tested using the Genepop on the Web (https://genepop.curtin.edu.au/ accessed on 8 June 2025) [52]. The within-population genetic characteristics, i.e., observed (*H_o_*) and expected (*H_e_*) heterozygosity, inbreeding coefficient (*F_IS_*), and the mean number of alleles (*A*), were computed in Genetix 4.05 [53]; allelic richness *A_R_* in Fstat 2.9.4 [54]; and the number of private alleles (*A_P_*) in GenAlEx 6.503 [55]. GenAlEx was also used to calculate fixation index *F_ST_*, and to perform analyses of molecular variance that evaluate the distribution of genetic diversity within and among populations (AMOVA). We performed separate AMOVAs for all populations, comparison among countries, comparison within PL and RO, and comparison of two generations in RO population Vad (cf. Table 1).

We assessed the isolation in terms of distance by linear regressions of pairwise *F_ST_* values against aerial distances (in km), in the form *F_ST_*_(i–j)_/(1 − *F_ST_*_(i–j)_)~log_10_(Distance_(i–j)_). To relate the within-population genetic parameters to characteristics of the sites, we calculated the Pearson’s correlation coefficients with the ordinal values of *area* and *connectivity.*

Neighbour-joining clustering based on within-population allelic frequencies was used to construct a dendrogram of the populations, using PAST 4.17 [56]. For more detailed assessment of relationships among populations, we used the Bayesian programme STRUCTURE 2.3.4 [57], applying no admixture model and independent allele frequencies, 10 independent simulations for each K = 1–10, with 500,000 Markov chain Monte Carlo (MCMC) repetitions after a 100,000 burn-in period. To combine the results of individual runs, we used StructureSelector (https://lmme.ac.cn/StructureSelector/ accessed 12 June 2025) [58], inferring the most likely number of clusters by combining the likelihood of the K selection method [59] and the Delta K method [60].

## 3. Results

The average number of individuals per population was 16.1 ± 6.28 SD, range 8–32. The maximum number, from RO locality Vad, consisted of two groups of 16 individuals each, sampled during spring and summer generations. The results from populations with lowest numbers of individuals (Straszewo PL, Šumava CZ) should be interpreted with caution (Table 1).

None of the pairs of loci were in significant linkage disequilibrium. On the other hand, only three populations (Šumava, Straszewo, and Coșna) did not deviate from the Hardy–Weinberg equilibrium, indicating presence of microevolutionary processes.

The values of expected heterozygosity *H_e_* were rather high and always higher than observed heterozygosity *H_o_* (paired t_(*n*=10)_ = 6.74, *p* < 0.001) (Table 1). They were higher in PL (mean 0.75 ± 0.066 SD) than in RO (0.58 ± 0.091 SD) (t_(7*df*)_ = 3.37, *p* < 0.05). The Šumava CZ heterozygosity values were identical to PL averages. The *H_o_* values were also higher in PL than in RO but did not differ statistically (0.55 ± 0.134 SD vs. 0.42 ± 0.108 SD, t_(7*df*)_ = 1.64, *p* = 0.14). The numbers of alleles per locus *A* and the numbers of private alleles *A_P_* were also quite high in average, both higher in PL than in RO, but not significantly so (*A*: 7.37 ± 2.267 SD vs. 5.10 ± 1.190, t_(7*df*)_ = 0.57, *p* = 0.16; *A_P_*: 8.60 ± 5.030 vs. 3.50 ± 2.380, t_(7*df*)_ = 1.85, *p* = 0.11). The higher numbers of alleles in PL were driven by the large and interconnected Polish population of Siedliszcze. The lowest numbers of alleles were in the isolated Romanian populations of Coșna (high altitude) and of Vad (low altitude).

Values of inbreeding coefficient *F_is_* were negatively correlated with *H_o_* (r_(*n*=10)_ = −0.78, t = −3.56, *p* < 0.01), but not with the other measures of genetic variability (all *p*’s > 0.50). The mean values were similar in PL and RO (0.274 ± 0.1481 vs. 0.288 ± 0.1151, t_(7*df*)_ = −0.16, *p* = 0.88) and ranged widely, from a negligible value in Straszewo PL (0.056) to highly inbred populations Bobry PL (0.465) and Vad RO (0.454).

Comparing the spring and summer generations samples from Vad gave very similar values of all the genetic diversity parameters, overlapping with values for the pooled sample.

A formal comparison of the introduced Šumava population with all remaining populations returned marginally significantly lower values for *H_e_* (single-term t_(8*df*)_ = 1.95, *p* = 0.09), *A* (t_(8*df*)_ = 3.98, *p* < 0.01) and *A_P_* (t_(8*df*)_ = 2.13, *p* = 0.07), but no differences in *H_o_* and *F_is_* (both *p*’s > 0.35). Also, the values of *H_e_*, *H_o_*, and *A* were lower than the values detected at the source locality in the Eastern Alps (Table 1).

Correlation analysis relating genetic parameters to the *area* of the sites (Table 2) did not yield significant results, but the values of correlation coefficients *r* were generally high and positive, implying higher genetic diversity in populations inhabiting larger areas. Correlations with *connectivity* revealed an identical pattern, plus the inbreeding coefficient *F_IS_* declined with *connectivity*, and allelic richness *A_R_* displayed a marginally significant increase along with it. Note that due to low number of sites, results of this correlation analysis only have explorative character.

The pairwise *F_ST_* values (mean 0.209 ± 0.0798 SD) were all significant, attesting to differentiation among the populations (Table 3). The highest values, >0.30, separated the introduced Šumava CZ population from the PL population Kijowiec and RO population Coșna; the PL population Podbiel from RO population Coșna; and the RO population Vad from PL population Kijowiec and RO population Coșna. The lowest values, <0.1, were between PL populations Siedliszcze and Straszewo, and between RO populations Lăpușel and Moldova Sulița. The RO population Vad was the most differentiated, followed by RO population Coșna and PL population Kijowiec. The PL population Siedliszcze was the least differentiated, followed by PL populations Bobry and Straszewo. Within countries, the pairwise values did not differ between PL and RO (means 0.148 ± 0.0603 vs. 0.196 ± 0.0733, t_(17*df*)_ = −1.52, *p* = 0.15). The pairwise comparison of data pooled for entire countries revealed the highest differentiation between the introduced CZ population vs. RO (*F_ST_* = 0.163), followed by CZ vs. PL (0.131) and PL vs. RO (0.092) (all *p* < 0.001).

The AMOVA results (Table 4) corroborated the above observations. Overall, one fifth of genetic variation was attributable to variation among populations, and similar value applied to the variation among populations within RO. The number decreased to 15% within PL, indicating that Polish populations were mutually less differentiated than Romanian populations.

Isolation by distance was significant across all populations (*b* = −0.130 ± 0.0588 SE, F_(1,43_ *_df_*_)_ = 4.92, *p* < 0.05, R^2^ = 0.08), and marginally significant across all populations, excluding Šumava (*b* = −0.125 ± 0.0687 SE, F_(1,34_ *_df_*_)_ = 3.29, *p* = 0.08, R^2^ = 0.09). It was not significant in PL only (*b* = −0.186 ± 0.1514 SE, F_(1,8_ *_df_*_)_ = 1.51, *p* = 0.25, R^2^ = 0.05) and RO only (*b* = 0.384 ± 0.4617, F_(*1,4*_ *_df_*_)_ = 0.69, *p* = 0.45, R^2^ = 0.15). Note that the PL and RO regressions are based on small sample sizes (n = 5, 4, respectively).

The dendrogram of allelic frequencies (Figure 2) revealed that besides the CZ Šumava population, the studied populations split into two branches. One contained the southeastern PL Siedliszcze population in a basal position, from which all the RO populations branched. Lowland and mountain populations alternated, and Lăpușel (lowland) + Moldova Sulita (mountain) were in terminal positions. The other branch contained the remaining PL populations, with the northeastern Straszewo in a basal position, and Podbiel + Kijowiec, both from East-Central Poland, in terminal positions.

The structure analysis suggested division of the populations into four (Delta K method) or eight (likelihood of the K) clusters; six clusters were also permissible according to Delta K (Figure 2). According to all three approaches, the lowland RO population Vad and mountain RO population Coșna are genetically uniform and differentiated from all others. The lowland Lăpușel and mountain Moldova Sulita are genetically diverse, with the latter being more so, and Moldova Sulita contains alleles from both Coșna and Lăpușel. In PL, the southeastern Siedliszcze population is genetically diverse, Kijowiec and Podbiel are grouped together under K = 4, 6, and Straszewo is grouped with Bobry under all three Ks. The introduced CZ population differs from all others under K = 6, 8, whereas under K = 4, it is most related to the lowland RO population Lăpușel.

## 4. Discussion

Exploring genetic diversity of populations of the continentally threatened *L. helle* butterfly in the lowlands (Poland) and mountains (Carpathian Arc of Romania) of East-Central Europe revealed multiple similarities, but also remarkable differences, with earlier results obtained mainly from Western and Central Europe. While within-population genetic parameters were within the previously reported ranges, we detected in both countries unexpected levels of population differentiation, even among localities not isolated by obvious geographic barriers in Poland and geographically quite proximate in the Carpathians. The introduced population in Šumava, CZ, originally from the Eastern Alps, lost some genetic diversity when compared with its donor population, but the resulting population genetic parameters were within the range of many extant and viable European populations.

### 4.1. Within-Population Patterns

The earlier study covering genetics of *L. helle* across Europe, with emphasis on Western and Central European mountains, included also three Polish lowland sample and one Romanian mountain sample, plus one lowland sample from Mecklenburg-Vorpommern (NE Germany) and one from Lithuania [33]. All these samples, when loosely grouped together with samples from the Eastern Alps, exhibited similar means and maxima of population genetic parameters to samples from mountains of Western Europe, but less prominent genetic differentiation. The most obvious explanation—a more contiguous distribution of habitats in Eastern Europe—would be valid only in a broad view. In closer detail, the Polish and Romanian populations vary considerably. The sample with the highest number of unique alleles is the large and well protected population Siedliszcze, which was also analysed, with nearly identical results, in [33]. We found the highest observed heterozygosity in the sample from Straszewo, representing the Podlasie region of northeastern Poland, and notable for diverse landscapes. The site is located within the Knyszyn Forest, where more than 20 local colonies of *L. helle* persist [61,62]. In contrast, the PL population Bobry, situated rather centrally in Poland, exhibited high inbreeding coefficient, indicative of isolation, and/or population size bottlenecks in a past. Two PL populations with intermediate values of genetic indices, Kijowiec and Podbiel, indicate that contrasting environmental pressures may bring similar outcomes. The Kijowiec population is strong and stable, although relatively isolated, whereas Podbiel, part of a wider metapopulation system, has decreased in abundance due to inappropriate management.

The four Romanian populations represent even higher diversity of genetic patterns, but also diverse habitat associations. The two lowland bivoltine populations, more specifically, the southernmost lowland *L. helle* populations in Europe, do not inhabit grasslands, but open-canopy oak-dominated forests with *B. officinalis* in understory (Figure 1). This habitat use resembles that of iconic open forest butterflies such as *Coenonympha hero*, *Euphydryas maturna*, *Lopinga achine*, or *Parnassius mnemosyne* [63,64,65,66]. This is a strong indication that although *L. helle* is considered as a mammoth steppe relict associated with grasslands, it can sustain itself in damp “temperate savannas”, the prevailing vegetation type in lowland Central Europe during interglacial periods [67]. This land cover was maintained in an open-canopy state by activities of large herbivores in the distant past [68], and by previous forest pasture and coppicing practices [69]. The highly isolated and genetically unique lowland population Vad is genetically impoverished by drift and inbreeding processes, and at the same time at risk of eventual canopy closure. As a strict nature reserve, Vad forest is not managed for open forest conditions, which may in the long term threaten not only *L. helle*, but also unique local flora, including several species of *Narcissus* spp. [70].

The Romanian mountain populations inhabit landscapes and habitats similar to those in the middle–high mountains of Western Europe [20,71] (Figure 1). They share these habitats with other glacial relic Lepidoptera, such as the bog fritillary *Boloria eunomia* [25,39,72], which uses the same host plant and also depends on non-intensive management of mountain grasslands, with temporary retention of successionally advanced patches [26,27,73]. Geographically, the mountain populations are much closer to the lowland Lăpușel population than to lowland Vad population, and their mutual aerial distance is just ≈30 km. Still, while the mountain Moldova Sulita population is only minimally differentiated from Lăpușel and displays similar genetic composition, the mountain Coșna population, despite rather high heterozygosity values, displays radically different genetic composition. The closest distance separating the two mountain populations is across the ridge of Rodna Mts. (≈2000 m altitude), a formidable barrier. It is also possible that lowland populations related to Coșna are already extinct.

The Šumava, CZ, population, established two decades ago by transfer of 2 males and 17 females (details [39]), lost 22.4% (*H_e_*) and 17.1% (*H_o_*) of heterozygosity of its donor Eastern Alps population. A higher loss appears to have affected the numbers of alleles per locus *A*, but this parameter depends on sample size, which was lower in this study than in the Eastern Alps sample by Habel et al. [41]. In any case, the genetic parameters of the introduced populations are within the limits of many populations assessed so far and the population was capable of expansion, currently inhabiting >60 ha of habitat, with current distribution limits >4 km from the original release point [39].

### 4.2. Conservation Biogeography in East-Central Europe

Despite the rather low number of populations sampled, and hence low n in correlation analysis, the larger area and higher connectivity of the populations tended to be associated with higher values of population diversity parameters, and with lower values of inbreeding coefficient. A better performance of connectivity than area was found in several butterfly population genetic studies (e.g., *Parnassius smintheus* [74], *Polyommatus coridon* [75], *Phengaris alcon* [76]) and is explainable by metapopulation processes. Large areas of habitat do not exclude fluctuation in population numbers, e.g., due to spatially synchronised effects of weather, or changing of habitat quality. Good connectivity, on the other hand, allows genetic rescue in case of local genetic diversity loss, as empirically demonstrated in a metapopulation of *Melitaea cinxia* [77].

The crucial role of population connectivity for genetic diversity is illustrated by lower population differentiation within PL than within RO (and Western Europe [28,33]), suggesting shorter mutual isolation within the former. The network of habitable sites in the Polish lowlands was certainly much denser prior to modern agricultural intensification. Wetlands, including humid grasslands, were historically affected by drainage followed by conversions to productive fodder meadows and arable fields, exacerbated by climate change [78,79,80]. The North European Plain does not contain many substantial barriers, which is mirrored by genetic structures of some butterfly species, whose genetic lineages form broad latitudinally distributed bands across Eurasia [3]. Although long distance dispersal is rare in *L. helle*, it certainly exists, as documented by observing translocated populations [39,44]. The bivoltine cycle of the lowland populations likely enhances the probability of reaching distant habitats relative to univoltine populations, given that the second generation, typically more abundant in PL, produces higher numbers of emigrating females [32]. The Romanian forest-dwelling bivoltine population Lăpușel, however, produced a less abundant second generation according to the mark-recapture study [19].

The highly differentiated forest-dwelling populations in Romania offer some insights onto glacial and postglacial dynamics of the species. Although the non-mountain populations of *L. helle* recently inhabit mainly northern temperate (i.e., Poland) and boreal vegetation belts, this was not the case during the glacial maximum, when much of North European Plain was under the glaciers. Habel et al. [33] used species distribution modelling to demonstrate that during that time, suitable habitats existed south of (approximately) 53 N parallel, in lowlands of what is now France, Belgium, southern Germany, the Czech Republic, southeastern Poland, and Romania. Contrasting with the populations in central and northern Poland, the outstandingly diverse southeastern PL population Siedliszcze, but also the diverse lowland Romanian population Lăpușel and the impoverished, but still genetically unique population Vad, have likely persisted the last glacial cycle in situ, and provided for postglacial colonisation of high elevations of the Carpathians. Genetic diversity losses during postglacial colonisation were previously described for other temperate-zone butterflies [81,82,83]. The diversification between populations in southeastern vs. central and northern Poland might be a result of postglacial colonisation of the more northerly regions from eastern unglaciated areas, as was shown, e.g., for the boreal forest butterfly *Coenomympha hero* [84] and several others [5]. To ascertain this, sampling of *L. helle* from more easterly regions would be necessary.

The genetic proximity of lowland bivoltine and mountain univoltine populations in Romania, and the closer relatedness between mountain and lowland populations than between lowland or between lowland populations, also implies that voltinism patterns may change rather rapidly and repeatedly. Rapid increases in butterflies’ generation numbers associated with warmer climates are currently observed from many regions [85,86], and applies even to lowland *L. helle* populations in northern Poland [50].

This returns us to the outstanding conservation value of the southeastern Polish and Romanian lowland populations. Together, they arguably represent a rear edge of postglacial *L. helle* expansion towards higher altitudes of the Carpathians (cf. Figure 2). The relatively high representation of private alleles, contrasting with intermediate (Lăpușel) and low (Vad) genetic diversity, echoes other cases of genetically impoverished, but uniquely adapted relic populations, described, e.g., in declining mammals [87,88]. Such populations raise a conservation dilemma, whether to strive for increased genetic diversity via connectivity enhancement, or whether to conserve their uniqueness via more efficient management of their habitats. The ability of these populations to persist through the warm Holocene period in situ indicates a broader thermal tolerance of *L. helle* species than usually assumed (the distribution models in [33] did not consider the Romanian localities for defining *L. helle* thermal requirements). The persistence of Romanian lowland populations in open forests illustrates how the butterfly had survived in temperate Europe during the period between retreats of glaciers, forest expansion, and forest clearance by early agriculturalists (cf. [89,90]). Youri et al. [91] called for future study of functional genetics in isolated populations of such relic species as *L. helle*. Due to their high allelic richness, these rear edge populations certainly deserve attention with respect to their thermal physiology, and other aspects of survival in extreme conditions.

## Figures and Tables

**Figure 1 insects-16-01202-f001:**
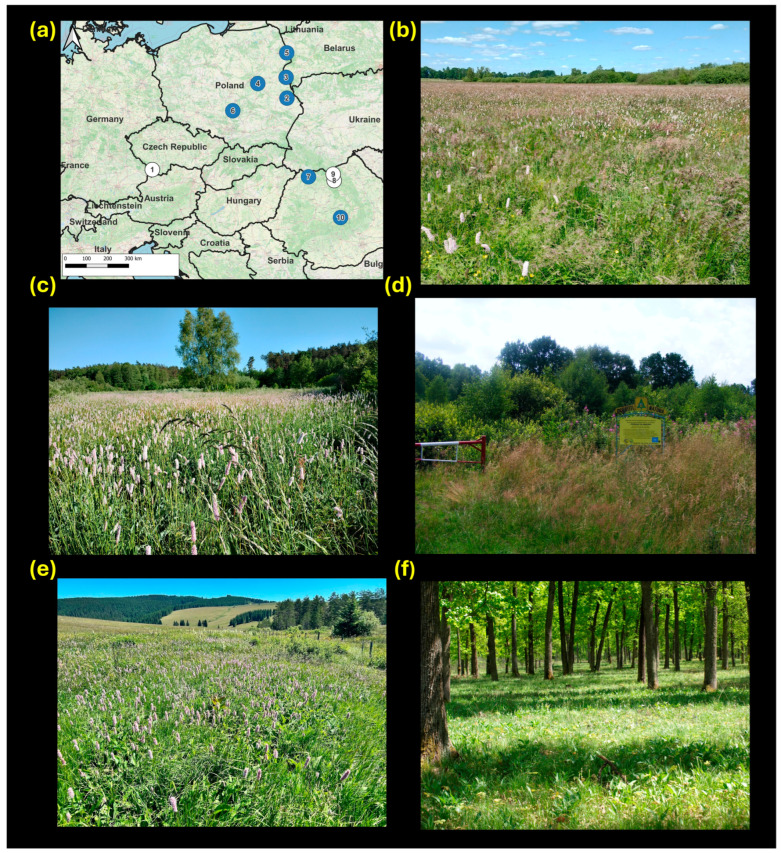
(**a**) Map of East-Central Europe with the sampled populations of the *Lycaena helle* butterfly, with the white circles indicating bivoltine/mountain populations and the blue circles showing lowland/univoltine populations. (**b**) Siedliszcze, PL (No. 2 in Table 1), an example of the large and non-isolated lowland population; (**c**) Kijowiec, PL (No. 3), an example of isolated lowland population; (**d**) Vad forest, RO (No. 10), a lowland bivoltine population, photographed during summer generation’s flight period; (**e**) Moldova Sulita, RO (No. 9), a high-elevated Carpathian population; and (**f**) Lăpușel forest, RO (No. 7), a lowland bivoltine population photographed in late spring.

**Figure 2 insects-16-01202-f002:**
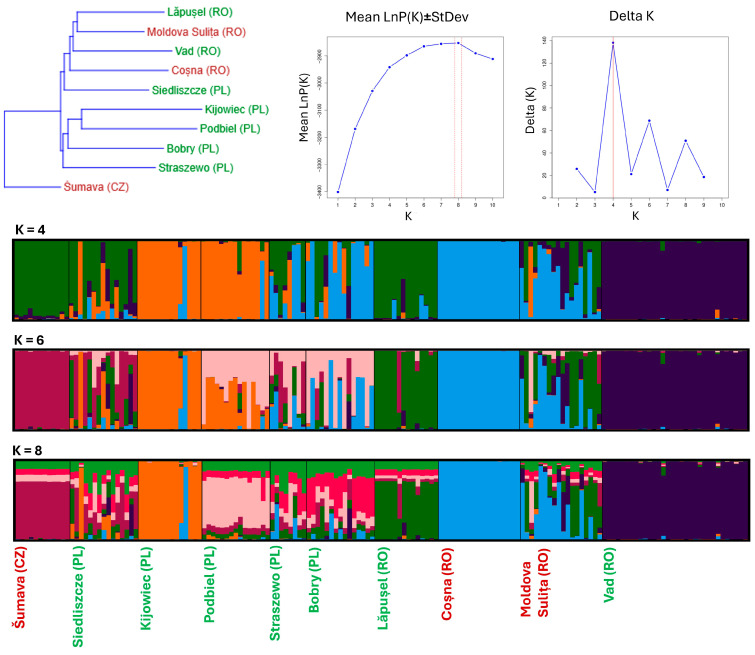
Results of analysis of relations among *Lycaena helle* populations in East-Central Europe. Top row shows a dendrogram based on allelic frequencies, and outputs of StructureSelector, indicating the right numbers of clusters according to probability of K and Delta K methods. The lower section shows the STRUCTRURE diagrams according to K = 4, 6, and 8. The locality names in green and brown indicate lowland and highland populations, respectively.

**Table 1 insects-16-01202-t001:** Overview of the *Lycaena helle* populations targeted by this study; their basic ecological characteristics, sample sizes, and population genetic parameters obtained by analysing microsatellite markers. The lines in *italics* were not considered during the calculations for overall sums or means. The ecological characteristics *Area* and *Connectivity* (=*Con.*) are at ordinal scales 1–3, altitude (Alt.) is in metres, voltinism (Volt.) distinguishes univoltine, U, or bivoltine, B, populations. Population genetic parameters are expected heterozygosity *H_e_*, observed heterozygosity *H_o_*, inbreeding coefficient *F_IS_*, the number of alleles per locus *A*, allelic richness *A_R_*, and the number of private alleles *A_P_*. *HWE* are results of tests examining deviation from the Hardy–Weinberg equilibrium (***: Bonferroni-corrected values < 0.001).

	ID	Site	*Area*	*Con.*	Alt.	Volt.	*N*	*H_E_* ± SE	*H_O_* ± SE	*F_IS_*	*A*	*A_R_*	*A_P_*	*HWE*
CZ	1	Šumava	3	1	860	U	12	0.60 ± 0.163	0.46 ± 0.406	0.248	3.3	n.a. ^d^	3	0.16
*AT*	*1a* ^a^	*Mariazell*	*2*	*1*	*900*	*U*	*30*	*0.77 ± n.a.*	*0.54 ± n.a.*	*n.a.*	*11.4*	*5.0*	*n.a.*	*–*
PL	2	Siedliszcze	3	3	200	B	15	0.83 ± 0.179	0.63 ± 0.382	0.247	11.4	4.4	17	***
PL	3	Kijowiec	1	1	150	B	14	0.68 ± 0.152	0.49 ± 0.181	0.286	6.2	3.2	7	***
PL	4	Podbiel	3	2	100	B	15	0.69 ± 0.198	0.48 ± 0.311	0.319	6.2	3.3	9	***
PL	5	Straszewo	2	2	150	B	8	0.79 ± 0.194	0.75 ± 0.468	0.054	6.3	n.a. ^d^	4	0.13
PL	6	Bobry	2	2	210	B	15	0.77 ± 0.152	0.40 ± 0.353	0.465	6.8	3.7	6	***
RO	7	Lăpușel	3	2	180	B	14	0.61 ± 0.363	0.45 ± 0.319	0.278	5.8	3.2	6	***
RO	8	Coșna	1	2	950	U	18	0.50 ± 0.286	0.41 ± 0.299	0.200	3.4	2.4	1	0.02
RO	9	Moldova Sulița	3	1	1230	U	16	0.69 ± 0.350	0.54 ± 0.311	0.221	7.2	3.6	5	***
RO	10	Vad	2	1	500	B	32	0.50 ± 0.293	0.28 ± 0.349	0.454	3.6	2.5	3	***
*RO*	*10a* ^b^	*Vad spring*					*16*	*0.41 ± 0.298*	*0.25 ± 0.354*	*0.423*	*3.0*			*0.12*
*RO*	*10b* ^b^	*Vad summer*					*16*	*0.55 ± 0.316*	*0.32 ± 0.350*	*0.463*	*3.4*			*0.04*
Sum, or mean ± SD ^c^							Σ 189	0.67 ± 0.113	0.49 ± 0.129	0.277 ± 0.1199	6.1 ± 2.42	3.3 ± 1.49	6.0 ± 4.55	

^a^ taken from Habel et al. [33]; ^b^ two samples from identical population, analysed as pooled sample in most of the analyses; ^c^ the values from Mariazell, and the two seasonal samples from Vad, not considered in sums and means; ^d^ value not computable due to small sample size.

**Table 2 insects-16-01202-t002:** Results of Pearson’s correlations relating population genetic parameters of *Lycaena helle* populations to the *area* and *connectivity* of the sites. All correlations are at 10 data points, except for *A_R_*, which is based on only 8 data points (cf. Table 1).

	*Area*			*Connectivity*		
Genetic Parameter	*r*	*t*	*p*	*r*	*t*	*p*
*H_E_*	0.36	1.03	0.34	0.49	1.49	0.18
*H_O_*	0.23	0.64	0.54	0.42	1.24	0.25
*F_is_*	0.02	0.06	0.96	−0.23	−0.61	0.56
*A*	0.56	1.77	0.12	0.54	1.69	0.14
*A_R_*	0.54	1.59	0.16	0.50	1.43	0.20
*A_P_*	0.52	1.62	0.15	0.63	2.12	0.07 ^#^

^#^: *p* < 0.1.

**Table 3 insects-16-01202-t003:** Pairwise *F_ST_* values and their significance levels (***: *p* < 0.001) for selected *Lycaena helle* populations in East-Central Europe, and mean *F_ST_* values for individual populations.

	Šumava	Siedliszcze	Kijowiec	Podbiel	Straszewo	Bobry	Lăpușel	Coșna	Moldova Sulița	Mean per Site ± SD
Šumava										0.239 ± 0.0682
Siedliszcze	0.169 ***									0.127 ± 0.0512
Kijowiec	0.315 ***	0.127 ***								0.245 ± 0.0687
Podbiel	0.227 ***	0.154 ***	0.222 ***							0.231 ± 0.0828
Straszewo	0.156 ***	0.099 ***	0.261 ***	0.115 ***						0.182 ± 0.0719
Bobry	0.200 ***	0.067 ***	0.194 ***	0.144 ***	0.104 ***					0.160 ± 0.0587
Lăpușel	0.240 ***	0.083 ***	0.241 ***	0.263 ***	0.173 ***	0.143 ***				0.191 ± 0.0697
Coșna	0.369 ***	0.170 ***	0.286 ***	0.351 ***	0.292 ***	0.201 ***	0.226 ***			0.266 ± 0.0786
Moldova Sulița	0.260 ***	0.066 ***	0.201 ***	0.261 ***	0.189 ***	0.126 ***	0.097 ***	0.163 ***		0.175 ± 0.0687
Vad	0.212 ***	0.208 ***	0.355 ***	0.339 ***	0.251 ***	0.258 ***	0.257 ***	0.335 ***	0.214 ***	0.270 ± 0.0582

**Table 4 insects-16-01202-t004:** Results of analysis of molecular variance among the studied populations of *Lycanea helle* butterfly in East-Central Europe.

Analysis (Number of Groups)	Variance	*F_ST_*	*F_IS_*	*F_IT_*
**All populations (n = 10)**	2.032	0.446	0.515	1.071
% variation, significance tests		21.9%, 9df ***	25.4%, 151df ***	52.7%, 161df ***
**Among countries (n = 3)**	2.081	0.107	0.423	0.485
% variation, significance tests		10.7%, 2df ***	37.8%, 158df ***	51.5%, 161df ***
**Within Poland (n = 5)**	2.162	0.150	0.322	0.424
% variation, significance tests		15.0%, 4df ***	27.3%, 62df ***	57.6%, 67df ***
**Within Romania (n = 4)**	1.855	0.230	0.334	0.487
% variation, significance tests		23.0%, 3df ***	25.7%, 78df ***	51.3%, 82df ***
**Vad among generations (n = 2)**	1.184	−0.012	0.432	0.426
% variation, significance tests		0.0%, 1df	43.2%, 30df ***	56.8%, 32df ***

*** *p* < 0.001.

## Data Availability

The table of microsatellite genotypes are provided in Appendix A of the article.

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
