# Peer review of "Eastern Arc of Glacial Relict Species—Population Genetics of Violet Copper Lycaena helle Butterfly in East-Central Europe"

_insects, 2025, doi:10.3390/insects16121202_

Round 1

Reviewer 1 Report

Comments and Suggestions for Authors

Review on the paper by Sitar et al. - A very interesting study, providing additional information on the population genetics and biogeography of L. helle, a boreo-montane butterfly with disjunct distribution across the Western Palaearctis. I recommend publication of this manuscript. I found just some few minor points for improvement (rather general structure, lack of focusing etc.) Please find my few comments in chronological order:

Title – I do not understand the title of the manuscript, what do you mean with “Eastern arc of glacial relict”? Also, the title might clearly describe the main focus of your study, this would help a lot.

Abstract:

Abstract needs clear focusing and a re-writing; I suggest to start the abstract with a general introduction, then your study-set-up, main findings, I suggest to avoid details such as values; a clear thematic focus.

This is also my main point in respect of this paper; what is the main focus of the paper? Is this study about the biogeography of the butterfly species? About potential genetic adaptations (lowland vs highland), or is it about the genetic effects from translocating individuals (from Austria to somewhere? Or is it really about all these three aspects; then it might be a bit a challenge to clearly focusing this work.

Introduction – in general ok, but also here there is a lack of clear focusing on a specific topic.

L78 continentally endangered … butterfly, rather: endangered butterfly with continental distribution ….

  1. helle – always in capitals

Material and Methods

In general well written, all is well explained; but I suggest to build up larger paragraphs (some paragraphs consist of just one single sentence);

Results – ok

Discussion

Well written, with many references; the authors are comparing their results very well with other work;
In my opinion it is very important to differentiate between effects from recent habitat fragmentation and patterns arising from the biogeographic history; This is not very clear in some sections.

Author Response

REVIEWER 1  

Title – I do not understand the title of the manuscript, what do you mean with “Eastern arc of glacial relict”? Also, the title might clearly describe the main focus of your study, this would help a lot.

### We agree that the title was a bit too poetic, little informative. In the present version, we change it to “Eastern ways of a glacial reclict relict – population genetics of the Violet Copper Lycaena helle butterfly in East-Central Europe”

Abstract:

Abstract needs clear focusing and a re-writing; I suggest to start the abstract with a general introduction, then your study-set-up, main findings, I suggest to avoid details such as values; a clear thematic focus.

### Although we understand your point, we cannot really follow it, because the Journal, as per Instructions for authors, requires rather specific Abstract structure, with a first part called “Simple summary (background and main message for laymen), and a second part with more technical results.

This is also my main point in respect of this paper; what is the main focus of the paper? Is this study about the biogeography of the butterfly species? About potential genetic adaptations (lowland vs highland), or is it about the genetic effects from translocating individuals (from Austria to somewhere? Or is it really about all these three aspects; then it might be a bit a challenge to clearly focusing this work.

 ### You are right, it is a bit challenging. The very point is, that L. helle was extraordinarily deeply studied across mountains of Western Europe; we are supplementing the western results by various aspects observerable in East-Central Europe, and as it happens, there are more such aspects, of similar weight and importance.

Introduction – in general ok, but also here there is a lack of clear focusing on a specific topic.

 ### We did our best to sharpen the focus. Perhaps the main change, recommended also by other Reviewers, was to shift the focus into the situation in Poland and Romania, and to explicitly state that the introduced Czech population is analysed only as a link with the situation in mountains of Western Europe, and does not represent the focal question of the paper.

L78 continentally endangered … butterfly, rather: endangered butterfly with continental distribution ….

### Not really, we really meant that it is endangered across the entire (European) continent.

helle – always in capitals

 ### Of course. We apologize for this omissions and did our best to correct them.

Material and Methods

In general well written, all is well explained; but I suggest to build up larger paragraphs (some paragraphs consist of just one single sentence);

### Following your advice, we rearranged the paragraphs a bit to get rid of those single-sentence paragraphs.

Results – ok

Discussion

Well written, with many references; the authors are comparing their results very well with other work.

In my opinion it is very important to differentiate between effects from recent habitat fragmentation and patterns arising from the biogeographic history; This is not very clear in some sections.

### Well, although we cannot agree more, drawing this distinction is sometimes difficult, as the history of many of the populations studied is not known yet. They are tractable, e.g. vias methods of historical landscape modelling, but not much tracked. Our attempt for this differentiating is in Discussion structure – separating within-population effects, presumably rather recent, and conservation biogeography, presumably much older. We hope that you will understand.

Reviewer 2 Report

Comments and Suggestions for Authors

Review of Sitar et al. “Eastern arc of a glacial relict - population genetics of the Violet Copper Lycaena helle butterfly in East-Central Europe”

Sitar and colleagues analyze population structure of protected palearctic insect. Based on microsatellite data, the authors found significant population structure at multiple spatial scales. The work highlights the distribution of genetic diversity and discusses conservation implications. Following revisions to improve communication, this manuscript would be appropriate for publication.

There are a few spots in the manuscript that could be improved in terms of the English language; to better communicate with the audience. A couple of examples:

Line 58: I think it should be “were established”, not “have established”.

Lines 79-82: “This humid grasslands’ dweller displays a wide Palearctic distribution, but only insular distribution in mountains of Western (from the Ardennes to Cantabrian Mts. and Pyrenees) and Central (Central Uplands and the Alps) Europe.” This sentence could be revised to something like: “The humid grasslands dweller is distributed broadly throughout the Palearctic, but is found in isolated populations in the mountains of Western and Central Europe.”

Lines 114-116: I don’t quite understand the question: “Did the Šumava transfer translated the bulk of the genetic variability of its source population in the Eastern Alps?”

Line 229: “but did no difference” should probably be “but did not differ”.

Line 347-348: I don’t understand “They host share with them other glacial relic species…” Are they feeding on the same host plant?

Paragraphs 1, 2, and 5 of section 4.2 could be significantly revised for readability.

Minor comments:

Line 45 (and elsewhere): Best to use univoltine throughout the manuscript, instead of monovoltine (both are used in the manuscript).

Line 94 (and elsewhere): L. helle should be italicized throughout the manuscript.

Line 105: Replace “ecological” with “ecologically”.

Lines 116-118: Add the word “does”: “How does the genetic make-up of populations…

Line 135: The sentence implies males are more restricted in their movements; if so, make this explicit.

Line 149: It would be worthwhile to remind readers what country abbreviations stand for at this point.

Line 151-152: Provide numeric ranges for this categories of area (e.g. 1: <10 ha; 2: 10-500 ha; 3: >500ha).

Line 154: How is a “well-connected metapopulation system” defined?

Line 185 (Table 1): Remove closing parentheses from superscripts.

Line 185 (Table 1): Allelic richness is not reported for two populations - is this because the sample sizes were too small to calculate AR? Another reason?

Line 204: The word “pair” should be pluralized: “pairs”.

Line 234: “larger populations” is an ambiguous term that could refer to the geographic area a population covers or the effective population size of a population. I think the authors are using it for the former, and may instead want to use “higher genetic diversity in geographically widespread populations” or something of that ilk.

Line 268: The superscript for non-significant comparisons could be omitted.

Line 270: Replace “significantly” with “significant”.

Line 274: Replace “low data points numbers” with “small sample sizes (n = 5, 4, respectively)”.

Line 334: Should P. bistorta instead be B. officinalis?

Line 433: It would be ideal if the data were made available in a more compute-friendly format, such as a Microsoft Excel or CSV file.

I look forward to reading a revised version of the work.

Sincerely,

Comments on the Quality of English Language

See comments in Comments & Suggestions for Authors section.

Author Response

REVIEWER 2

Review of Sitar et al. “Eastern arc of a glacial relict - population genetics of the Violet Copper Lycaena helle butterfly in East-Central Europe”

There are a few spots in the manuscript that could be improved in terms of the English language; to better communicate with the audience. A couple of examples:

Line 58: I think it should be “were established”, not “have established”.

Lines 79-82: “This humid grasslands’ dweller displays a wide Palearctic distribution, but only insular distribution in mountains of Western (from the Ardennes to Cantabrian Mts. and Pyrenees) and Central (Central Uplands and the Alps) Europe.” This sentence could be revised to something like: “The humid grasslands dweller is distributed broadly throughout the Palearctic, but is found in isolated populations in the mountains of Western and Central Europe.”

### Changed as recommended, see lines 79-81.

Lines 114-116: I don’t quite understand the question: “Did the Šumava transfer translated the bulk of the genetic variability of its source population in the Eastern Alps?”

### We apologize for the embarrassing grammar here. Changed to “Did the Šumava transfer translate the entire genetic variability of its source Eastern Alps population?”

Line 229: “but did no difference” should probably be “but did not differ”.

### After correcting the entire sentence structure, it is, in fact, “…but no differences in Ho and Fis“, see line 236.

Line 347-348: I don’t understand “They host share with them other glacial relic species…” Are they feeding on the same host plant?

### We apologise for this awkward wording, and changed it as follows: „They share the habitats with other glacial relic species, such as the bog fritillary Boloria eunomia [25, 39, 72], which uses identical host plant and also depends on non-intensive management of mountain grasslands, with temporary retention of successionally advanced patches [26, 27, 73]“

Paragraphs 1, 2, and 5 of section 4.2 could be significantly revised for readability.

 ### We did our best to smooth it.

Minor comments:

Line 45 (and elsewhere): Best to use univoltine throughout the manuscript, instead of monovoltine (both are used in the manuscript).

### checked and changed to “univoltine” across entire manuscript, including Table 1.

Line 94 (and elsewhere): L. helle should be italicized throughout the manuscript.

### Checked and corrected.

Line 105: Replace “ecological” with “ecologically”.

### Corrected as advised.

Lines 116-118: Add the word “does”: “How does the genetic make-up of populations…

### Done as suggested.

Line 135: The sentence implies males are more restricted in their movements; if so, make this explicit.

### Changed as suggested.

Line 149: It would be worthwhile to remind readers what country abbreviations stand for at this point.

### Done as suggested.

Line 151-152: Provide numeric ranges for this categories of area (e.g. 1: <10 ha; 2: 10-500 ha; 3: >500ha).

###Although this can be done rather easily, these exact numeric values imply that we actually measured the areas, which admittedly is not the case, because of troubles with transient zones present at each of the locality. We therefore prefer to leave it in the original state.

Line 154: How is a “well-connected metapopulation system” defined?

### This is similarly tricky matter, but we provided a definition, see line 163-164.

Line 185 (Table 1): Remove closing parentheses from superscripts.

### Done as suggested.

Line 185 (Table 1): Allelic richness is not reported for two populations - is this because the sample sizes were too small to calculate AR? Another reason?

###Exactly so, we added the explanation to the table, under the superscript “d”.

Line 204: The word “pair” should be pluralized: “pairs”.

###Corrected as recommended.

Line 234: “larger populations” is an ambiguous term that could refer to the geographic area a population covers or the effective population size of a population. I think the authors are using it for the former, and may instead want to use “higher genetic diversity in geographically widespread populations” or something of that ilk.

### We changed this to “populations inhabiting larger areas”, as this is more precise (none of the populations was truly widespread, all are relic and spatially limited to some extend).

Line 268: The superscript for non-significant comparisons could be omitted.

### Done as suggested.

Line 270: Replace “significantly” with “significant”.

###Corrected as suggested.

Line 274: Replace “low data points numbers” with “small sample sizes (n = 5, 4, respectively)”.

###Done as recommended.

Line 334: Should P. bistorta instead be B. officinalis?

###You are right, of course. We changed it all to Bistorta officinalis.

Line 433: It would be ideal if the data were made available in a more compute-friendly format, such as a Microsoft Excel or CSV file.

 ### We are providing excell file to this version.

Reviewer 3 Report

Comments and Suggestions for Authors

Review recommendation of the manuscript titled “Eastern arc of a glacial relict — population genetics of the Violet Copper Lycaena helle butterfly in East-Central Europe” submitted to Insect.

1. Introduction

The introduction provides a good general background but could be more focused. It would benefit from a clearer statement of the specific knowledge gaps in East-Central Europe that this study aims to fill. The final paragraph of the introduction states the questions, but the lead-up could more directly frame why these questions are critical for understanding the species’ biogeography and conservation in this particular region.

2. Experiment design and methods

Sampling rationale and justification: The rationale for selecting these specific ten populations is not sufficiently explained. The reader is left wondering why these particular sites in Poland and Romania were chosen over others. A brief justification based on their representation of different habitat types, voltinism, or conservation status, would strengthen the methodological foundation.

Sample size: The sample size for some populations is low (e.g., 8 for Straszewo, 12 for Šumava). The authors should explicitly discuss the potential limitations it imposes on the accuracy of population genetics, especially when making comparisons. A rarefaction analysis or the use of sample-size-corrected measures for all populations would be more appropriate for direct comparison.

Microsatellite markers: The use of only five microsatellite loci is lower than what is typically used for population genetic studies. The authors should provide a justification for this number or cite previous work demonstrating that this panel is sufficient to resolve population structure in this species. A mention of the markers’ polymorphism information content would be useful.

Statistical analysis:

(1) The sentence “We computed used Pearson’s correlations” is grammatically incorrect, please change to “We calculated the Pearson’s correlation coefficients”.

(2) For the correlation analysis between genetic parameters and area/connectivity (Table 2), the statistical power is likely very low. The authors should be more cautious in their interpretation, framing them as exploratory trends rather than definitive results. The marginally significant result (p=0.07) should not be highlighted without a strong caveat about multiple testing and low power.

(3) The AMOVA result (Table 4) is confusing. The headers Variance, Fst, Fis, Fit do not align clearly with the data rows below, which present percentages of variation. This table needs to be reformatted for clarity.

3. Presentation

Clarity and flow: The results section is dense and at times difficult to follow. The text reports multiple statistical comparisons in a single sentence, which can be hard to parse. Breaking down complex findings into simpler, more focused statements would improve readability.

(1) Table 1: In Table 1, “na” (not available) is used for some values without explanation. The methodology should be consistent across all populations, and reasons for missing data must be stated. The format of this table requires revision.

(2) Figure 2: The description of the STRUCTURE results (K=4, 6, 8) is somewhat confusing. The text jumps between different K-values, making it hard for the reader to grasp the primary, most supported pattern. The authors should first clearly state which K-value is best supported by their analysis, and then describe the predominant pattern for that K-value before discussing alternatives. The caption for Figure 2 is also too brief and does not adequately explain what is being shown in the different panels.

4. Discussion

Over-interpretation of patterns: The discussion sometimes over-interprets patterns based on limited data. The detailed speculation on the colonisation routes of northern Polish populations from “more easterly regions” is not strongly supported by the presented data.

Linking genetics to ecology and conservation: The discussion of the Romanian forest-dwelling populations is interesting but could be deepened. The authors state these populations “deserve the highest conservation priority”, but the argument would be stronger if directly linked to the genetic results.

Contradictory statements: There is a potential contradiction that should be addressed. The authors state that connectivity is crucial for genetic diversity, yet the highly isolated and inbred population possesses several private alleles. This illustrates how drift in isolated populations can create uniqueness, which presents a conservation dilemma. This tension between inbreeding depression and the value of unique genetic variants should be discussed.

Weakened conclusion on voltinism: The conclusion that “voltinism patterns may change rather rapidly” is based on genetic proximity of populations with different voltinism, but this is an indirect inference. The authors should tune down this claim or support it with references to contemporary observations of voltinism shifts due to climate change.

5. Logic and clarity

Grammar issues: The manuscript requires thorough proofreading for grammatical errors, awkward phrasing, and typos.

Logical issues: The flow between paragraphs, particularly in the discussion, can be improved. The narrative jumps between different topics without smooth transitions. Restructuring paragraphs to group related ideas would enhance coherence.

Comments on the Quality of English Language

The manuscript requires thorough proofreading and language edits by a native English speaker for grammatical errors, awkward phrasing, and typos.

Author Response

REVIEWER 3

  1. Introduction

The introduction provides a good general background but could be more focused. It would benefit from a clearer statement of the specific knowledge gaps in East-Central Europe that this study aims to fill. The final paragraph of the introduction states the questions, but the lead-up could more directly frame why these questions are critical for understanding the species’ biogeography and conservation in this particular region.

  1. Experiment design and methods

Sampling rationale and justification: The rationale for selecting these specific ten populations is not sufficiently explained. The reader is left wondering why these particular sites in Poland and Romania were chosen over others. A brief justification based on their representation of different habitat types, voltinism, or conservation status, would strengthen the methodological foundation.

### We added the rationale to section 2.2., lines 154-156.

Sample size: The sample size for some populations is low (e.g., 8 for Straszewo, 12 for Šumava). The authors should explicitly discuss the potential limitations it imposes on the accuracy of population genetics, especially when making comparisons. A rarefaction analysis or the use of sample-size-corrected measures for all populations would be more appropriate for direct comparison.

### We understand the problem, acknowledged it at lines 201-203, and toned down some of the comparisons throughout.

Microsatellite markers: The use of only five microsatellite loci is lower than what is typically used for population genetic studies. The authors should provide a justification for this number or cite previous work demonstrating that this panel is sufficient to resolve population structure in this species. A mention of the markers’ polymorphism information content would be useful.

### Although the number of loci is low, they are exactly the same primers used in earlier studies by J-C Habel and colleagues in Western Europe, and we opted to use precisely the same markers to facilitate direct comparisons. We emphasized this in the present version, lines 165-6.

Statistical analysis:

(1) The sentence “We computed used Pearson’s correlations” is grammatically incorrect, please change to “We calculated the Pearson’s correlation coefficients”.

### Corrected as suggested, now at lines 187-9.

(2) For the correlation analysis between genetic parameters and area/connectivity (Table 2), the statistical power is likely very low. The authors should be more cautious in their interpretation, framing them as exploratory trends rather than definitive results. The marginally significant result (p=0.07) should not be highlighted without a strong caveat about multiple testing and low power.

### We absolutely agree – the whole analysis was meant as only an exploratory exercise from the beginning. We now added this cautionary sentence at lines 245-6.

(3) The AMOVA result (Table 4) is confusing. The headers Variance, Fst, Fis, Fit do not align clearly with the data rows below, which present percentages of variation. This table needs to be reformatted for clarity.

### For this version, we did some minor realignments of the table to prevent this confusion.

  1. Presentation

Clarity and flow: The results section is dense and at times difficult to follow. The text reports multiple statistical comparisons in a single sentence, which can be hard to parse. Breaking down complex findings into simpler, more focused statements would improve readability.

### As suggested, we broke the long sentences and thus slowed down the pace of the text, e.g., at lines 210-211, 219-217.

(1) Table 1: In Table 1, “na” (not available) is used for some values without explanation. The methodology should be consistent across all populations, and reasons for missing data must be stated. The format of this table requires revision.

### Those “na” were indeed due to low sample sizes, we explained it somewhere above in this letter, and also in the legend of the Table.

(2) Figure 2: The description of the STRUCTURE results (K=4, 6, 8) is somewhat confusing. The text jumps between different K-values, making it hard for the reader to grasp the primary, most supported pattern. The authors should first clearly state which K-value is best supported by their analysis, and then describe the predominant pattern for that K-value before discussing alternatives. The caption for Figure 2 is also too brief and does not adequately explain what is being shown in the different panels.

### Here, it was not easy to select a most supported pattern, as both methods of K selection are suggesting different results. Fortunately, the main pattern is retained under all three Ks, the only major difference being the position of CZ Sumava population (related to Lapusel under K = 4, completely unrelated under K = 6,8). We did some rearrangements of the text, but without radical changes.

  1. Discussion

Over-interpretation of patterns: The discussion sometimes over-interprets patterns based on limited data. The detailed speculation on the colonisation routes of northern Polish populations from “more easterly regions” is not strongly supported by the presented data.

### We agree that given the data, the support for colonisation from the East is rather weak. There are strong analogies, however, in other butterfly models, which we now emphasise at lines 420-424.

Linking genetics to ecology and conservation: The discussion of the Romanian forest-dwelling populations is interesting but could be deepened. The authors state these populations “deserve the highest conservation priority”, but the argument would be stronger if directly linked to the genetic results.

### Linking the conservation to genetic results is straightforward – both Romanian woodland populations contain unique alleles, one is more and the other less inbred, the less inbred population is more heterozygous. Still, repeating all these details in the relevant part of Discussion would complicate the flow of the text, and so we mentioned all these facts, but without going into detail. See lines 416-17.

Contradictory statements: There is a potential contradiction that should be addressed. The authors state that connectivity is crucial for genetic diversity, yet the highly isolated and inbred population possesses several private alleles. This illustrates how drift in isolated populations can create uniqueness, which presents a conservation dilemma. This tension between inbreeding depression and the value of unique genetic variants should be discussed.

### You are right that this is fascinating topic, which seems to be little explored in butterflies, or insects conservation in general. We mention it in some detail at lines 433-437 (including the notorious Florida panther example), but fully acknowledge that this topic deserves deeper exploration.

Weakened conclusion on voltinism: The conclusion that “voltinism patterns may change rather rapidly” is based on genetic proximity of populations with different voltinism, but this is an indirect inference. The authors should tune down this claim or support it with references to contemporary observations of voltinism shifts due to climate change.

### We agree that the inference is indirect, which is, we believe, sufficiently expressed by our choice of verb (“implies” instead of “documents”, “shows”, etc.). The supportive evidence of recent shifts is there – references 85, 86. We believe that it is enough as a selection, as presenting more of recently published evidence would expand the list of references intolerably.

  1. Logic and clarity

Grammar issues: The manuscript requires thorough proofreading for grammatical errors, awkward phrasing, and typos.

Logical issues: The flow between paragraphs, particularly in the discussion, can be improved. The narrative jumps between different topics without smooth transitions. Restructuring paragraphs to group related ideas would enhance coherence.

### We tried, really. May be, the imperfection is due to our effort to differentiate local population genetics and biogeography/history. Hopefully, the current version is more coherent, though it certainly still can be improved.

Reviewer 4 Report

Comments and Suggestions for Authors

Abstract :

Line 41. As it is a fist mention of this population better to write “A transferred CZ population from Western Europe…..”

As you are focusing on Eastern populations, you could start with the Polish populations, then move on to those in Romania, and finally say that you are going to compare them (as an external group) with those in the Czech Republic (from Western Europe).

Introduction:

Line 80 to 85 please provide a map (or maps) including original distribution and actual one.

Line 94 italic for L. helle

Line 99 /100 Please note that these studies were conducted using microsatellites in order to compare them with COI, which is completely ineffective for such recent radiation.

Line 103 L. helle italic

Line 105 The introduction into Forez was somewhat successful, while the introduction into Morvan was an almost total failure, with only a few residual populations surviving. What is the situation in the Czech Republic?

End of introduction: I think your main questions mainly concern Eastern Europe (Poland and Romania). The Czech Republic is ultimately only a peripheral part of this work. In my opinion, it simply serves to illustrate the possibilities of population transfers for this species. So perhaps you should rearrange the order of your questions.

Material Methods:

May be to indicate in Fig.1 a) which populations are monovoltine (I use univoltine…please check which is the correct writing) and which are bi/trivoltine (different colors may be?)

Presentation of results:

I find your presentation a little confusing. I suggest separating the low-altitude populations, the mountain populations and those in CZ, then adding a comparative paragraph at the end of the results (as you did with Table 4, but based more on altitude). The fact is that, of course, the location names are obvious to you, but not to the reader, and I found it difficult to follow your text, having to constantly refer back to Table 1 and Figure 1.

Caption of Fig.2 remove ycaena and just keep L. helle.

Your dendrogram could be more explicit with colour codes for altitude and/or countries. The same applies to your structure diagrams, where your localities are mixed without any apparent logic. In addition, for some, the genotypes appear to be homogeneous, while for others there are mixtures. I did not find any clear mention of this point in the results and discussion.

Line 288 and 290 try to use ∆ rather than “Delta”

Discussion:

Line 343 Are there any programmes to introduce/ reinforce populations of large mammals in the Vad forest in order to balance this “open forest”?

Line 371 Just a detail but it seems that….finally (after many fights) the correct name came back to P. phoebus.

Line 384 In my experience, most individuals have a maximum dispersal radius of 800 m/1 km. However, older females are sometimes visible up to 4–5 km from their original location (in agreement with Piotr's results).

General comments: This article is interesting, but I find the results difficult to read. I think the story of CZ should be minimised, as it is not ultimately the focus of this study. The main effort should therefore be to improve the clarity of the text, but the results and conclusions are correct.

Author Response

REVIEWER 4

Abstract :

Line 41. As it is a fist mention of this population better to write “A transferred CZ population from Western Europe…..”

### Changed as suggested.

As you are focusing on Eastern populations, you could start with the Polish populations, then move on to those in Romania, and finally say that you are going to compare them (as an external group) with those in the Czech Republic (from Western Europe).

### You are right here, we changed the order of countries here, and throughout.

Introduction:

Line 80 to 85 please provide a map (or maps) including original distribution and actual one.

### With a big sorry, this would go far beyond the possibilities of the paper. Neither Habel and colleagues provided a map that would combine historical and recent distribution for NW Europe, where it would be easiest task, owing to high quality historical distribution atlases. Expanding beyond that region, it would get increasingly tricky: the available historical atlases have too coarser grid, or too coarse time periods. But you are absolutely right, that this exercise might be useful for a future publication.

Line 94 italic for L. helle

### We did our best to correct this embarrassing mistake throughout the manuscript.

Line 99 /100 Please note that these studies were conducted using microsatellites in order to compare them with COI, which is completely ineffective for such recent radiation.

### This is not entirely true – the main reason to use microsatellites was not because COI is ineffective, but because the microsats had been developed, and previously applied in Western Europe, facilitating immediate comparisons. The ineffectiveness of the mitochondrial COI marker is mentioned elsewhere (line 101-102) and it is not necessary to mention it again here.

Line 103 L. helle italic

### Thank you.

Line 105 The introduction into Forez was somewhat successful, while the introduction into Morvan was an almost total failure, with only a few residual populations surviving. What is the situation in the Czech Republic?

### In Descimon and Balechard (2014), the Morvan introduction is still presented as rather successful – rapid expansion across wide area, quite a high number of local colonies. The fact is that the publication is more than decade old. This is not the main focus of our paper, however, and hence, we only mentioned it passim, without going into much detail. The Czech introduction is described in detail in Peskarova et al. (2024) – it is easily detectable publication in Nota, and again, we prefer not going into too much detail.

End of introduction: I think your main questions mainly concern Eastern Europe (Poland and Romania). The Czech Republic is ultimately only a peripheral part of this work. In my opinion, it simply serves to illustrate the possibilities of population transfers for this species. So perhaps you should rearrange the order of your questions.

### Rearranged as suggested.

Material Methods:

May be to indicate in Fig.1 a) which populations are monovoltine (I use univoltine…please check which is the correct writing) and which are bi/trivoltine (different colors may be?)

### Following your advice, we differentiated the univoltine and bivoltine populations in the Map. Bi- vs trivoltine we cannot responsibly distinguish, as appearance of third generations seems to vary among years.

Presentation of results:

I find your presentation a little confusing. I suggest separating the low-altitude populations, the mountain populations and those in CZ, then adding a comparative paragraph at the end of the results (as you did with Table 4, but based more on altitude). The fact is that, of course, the location names are obvious to you, but not to the reader, and I found it difficult to follow your text, having to constantly refer back to Table 1 and Figure 1.

Caption of Fig.2 remove Lycaena and just keep L. helle.

Your dendrogram could be more explicit with colour codes for altitude and/or countries. The same applies to your structure diagrams, where your localities are mixed without any apparent logic.

### Following the suggestion, we re-worked the figure.

In addition, for some, the genotypes appear to be homogeneous, while for others there are mixtures. I did not find any clear mention of this point in the results and discussion.

### It is now mentioned explicitly in the paragraph dealing with Structure, i.e. lines 299-308.

Line 288 and 290 try to use ∆ rather than “Delta”

### Done across the paper.

Discussion:

Line 343 Are there any programmes to introduce/ reinforce populations of large mammals in the Vad forest in order to balance this “open forest”?

### Not any of which we would be aware. Generally, the whole topic is fascinating, but would go beyond the scope of the paper, so we only cited a local paper (citation 70: Székely and Görbe 2019) and hope that a deeper discussion on management of the locality will be organised in a future.

Line 371 Just a detail but it seems that….finally (after many fights) the correct name came back to P. phoebus.

### You are absolutely right here, but this raises the issue, how to cite relatively recent papers that are using freshly obsolete names. Mentioning there the currently valid P. phoebus would require additional reference. So, we leave P. smintheus there for this moment, and let this issue to editor’s discretion.

Line 384 In my experience, most individuals have a maximum dispersal radius of 800 m/1 km. However, older females are sometimes visible up to 4–5 km from their original location (in agreement with Piotr's results).

### This is very interesting point, which supports the observations of expansion rate of the introduced population (Peskarova et al., 2024, and the French experiences). Unfortunately, we cannot cite “comments by and anonymous referee”, but we would love to, and observations of such stray individuals provide strong support for existence of rare long distance dispersal, even in presumably sedentary species.

General comments: This article is interesting, but I find the results difficult to read. I think the story of CZ should be minimised, as it is not ultimately the focus of this study. The main effort should therefore be to improve the clarity of the text, but the results and conclusions are correct.

### We believe that the CZ story is now minimised enough, and that the focus is much more to lowlands of Poland, and Carpathian heights.

Round 2

Reviewer 3 Report

Comments and Suggestions for Authors

The revision addressed all raised questions and improved the manuscript up to par. I think it can be accepted for publication now. 

Comments on the Quality of English Language

The manuscript requires thorough proofreading and language edits by a native English speaker for grammatical errors, awkward phrasing, and typos.